# Toddlers' Active Gaze Behavior Supports Self-Supervised Object Learning

## Abstract

Toddlers learn to recognize objects from different viewpoints with almost no su-
pervision. Recent works argue that toddlers develop this ability by mapping close-
in-time visual inputs to similar representations while interacting with objects.
High acuity vision is only available in the central visual field, which may explain
why toddlers (much like adults) constantly move around their gaze during such in-
teractions. It is unclear whether/how much toddlers curate their visual experience
through these eye movements to support their learning of object representations.
In this work, we explore whether a bio-inspired visual learning model can harness
toddlers' gaze behavior during a play session to develop view-invariant object
recognition. Exploiting head-mounted eye tracking during dyadic play, we simu-
late toddlers' central visual field experience by cropping image regions centered
on the gaze location. This visual stream feeds time-based self-supervised learn-
ing algorithms. Our experiments demonstrate that toddlers' gaze strategy supports
the learning of invariant object representations. Our analysis also reveals that the
limited size of the central visual field where acuity is high is crucial for this. We
further find that toddlers' visual experience elicits more robust representations
compared to adults', mostly because toddlers look at objects they hold themselves
for longer bouts. Overall, our work reveals how toddlers' gaze behavior supports
self-supervised learning of view-invariant object recognition.

## 1 Introduction

Toddlers learn visual representations that support the recognition of object instances observed from
different viewpoints within their first year of life (Kraebel & Gerhardstein, 2006; Ayzenberg &
Behrmann, 2024). This early emergence of view-invariant recognition and the ease with which
adults perform this skill hide the complexities of learning it. Images reaching the retina vary dras-
tically when objects are turned in depth. Even state-of-the-art machine learning methods still make
absurd recognition mistakes when faced with unusual viewpoints of objects (Dong et al., 2022; Ab-
bas & Deny, 2023; Ruan et al., 2023). This raises the question of what learning mechanisms support
such view-invariant recognition in humans.

One of the main theories posits that the development of view-invariant object recognition rests on
the brain's ability to construct visual representations that slowly change over time (Földiák, 1991;
Li & DiCarlo, 2008; Miyashita, 1988). The main idea is that learners abundantly manipulate (or
walk around) objects while watching them, giving access to different views of a single object over
a short period of time. By learning slowly changing representations, a learner discards rapidly
changing information from an image (here, information about the view) and naturally builds view-
invariant representations. Following this idea, recent computational studies proposed to simulate
humans' visual experience by generating or curating large-scale temporal sequences of rotating
objects (Aubret et al., 2022; Schneider et al., 2021; Yu et al., 2023); they confirm that learning
slowly changing representations induces view-invariant object recognition. However, it is currently
unclear if and how a toddler's actual gaze behavior supports this learning mechanism.

The significance of active gaze behavior stems from the limited area of high-acuity vision in hu-
mans. This area, known as the central visual field, covers only a few degrees of visual angle, but it
dominates the extraction of semantic information in brain regions responsible for object recognition
(Quaia & Krauzlis, 2024; Yu et al., 2015). However, such a small area of the visual field may be se-

mantically unstable over time, as humans make three saccades per second on average. Then again, toddlers curate their own visual experience; compared to adults, objects held by toddlers appear bigger in the field of view due to their shorter arms (Bambach et al., 2018), select simpler stimuli (Anderson et al., 2024) and their visual inputs semantically change on a slower timescale (Sheybani et al., 2023). The latter point may be critical to make slowness-based learning operational.

In this paper, we explore whether a bio-inspired model of visual learning can utilize the actual eye-tracking derived visual experience of toddlers to develop invariant object representations. For this, we leverage a dataset of head-camera recordings and gaze tracking from toddlers and adults during play sessions (Bambach et al., 2018). To simulate central visual experience, we crop image patches centered on tracked gaze locations. Then, we train previously introduced time-based self-supervised learning (SSL) models (Schneider et al., 2021). Our analysis shows that: a) toddlers' gaze strategy boosts visual learning in comparison to several baselines; b) restricting learning to input from the central visual field improves object representations; and c) visual input from toddlers yields better representations than that from adults, which may be explained by toddlers looking longer at objects while manipulating them. In sum, our main contributions are:

- We present the first ever study training SSL models on natural egocentric visual input derived from eye tracking in toddlers during play sessions.
- We find that toddlers' gaze strategy improves the learning of invariant object representations compared to several baselines.
- We show that toddlers' visual experience is more suitable for learning object representations through time-based SSL than adults'.

## 2 RELATED WORK

**Computational studies of visual learning with temporal slowness.** Early computational studies found that slowness-based learning can extract representations of simple patterns that are invariant to position, size and rotation (Földiák, 1991; Wiskott & Sejnowski, 2002). Other works applied this principle to learn view-invariant object recognition (Wallis & Baddeley, 1997; Franzius et al., 2011; Einhäuser et al., 2005; Stringer et al., 2006). Recent advances in SSL allowed to scale the principle of temporal slowness to large sets of uncurated images of objects (Parthasarathy et al., 2022; Aubret et al., 2022; Schneider et al., 2021). This method was called SSL through time (SSLTT) (Aubret et al., 2022). On the machine learning side, SSLTT can boost category recognition (Aubret et al., 2024b; 2022; Sanyal et al., 2023), view-invariant object instance recognition (Schneider et al., 2021) and the alignment with human representations (Parthasarathy et al., 2023). On the cognitive modeling side, SSLTT can shape human-like inter-object semantic similarities (Aubret et al., 2024a) and combines well with visuo-language SSL to model object learning during dyadic play (Schaumlöffel et al., 2023). However, all these approaches use curated, synthetic, or third-person data, leaving unclear whether the statistical structure of toddlers' actual visual experience, combined with temporal slowness, can indeed support object recognition. Another notable work studied the learning of view-invariant object representations in impoverished visual environments through the eyes of young chickens in a controlled rearing experiment (Pandey et al., 2024). In contrast, we apply SSLTT on natural visual inputs extracted from head cameras carried by toddlers and/or adults during play sessions.

**Learning from egocentric videos.** There is a recent surge in trained machine learning models on egocentric video datasets, including models of temporal slowness. For instance, the large-scale Ego4d dataset (Grauman et al., 2022) has been used for training vision models (Nair et al., 2022; Ma et al.; Anderson et al., 2022). However, egocentric videos for toddlers have been missing (Anderson et al., 2022); this is a problem since existing research has found that the specific statistical structure of toddlers' visual experience supports their learning (Sheybani et al., 2024; Bambach et al., 2017; Sheybani et al., 2023). The SAYcam dataset presents longitudinal recordings of 150 hours (on average) from each of the three participating children (Sullivan et al., 2021). With SAYcam, computational studies have shown that SSL methods can learn category recognition, with/without temporal slowness (Orhan et al., 2020; Orhan & Lake, 2024; Orhan et al., 2024). Another related work studies whether the temporal and developmental structure of toddlers' visual experience supports category and action recognition, through temporal slowness (Sheybani et al., 2024). Yet, these computational

studies neglect the gaze location and its associated behavioral strategy, as their datasets do not include the precise location of the individual's gaze. We show in Section 4.2 that this is critical for learning good object representations.

**Gaze-aware representation learning.** Our work extends previous approaches that also leverage the gaze location of a human to train vision models (Bambach et al., 2016; 2018). They also compare the quality of representations trained with toddlers' versus adults' experiences. However, these studies model the learning process with supervised learning, which is biologically implausible. This is important as, unlike bio-inspired self-supervised models that learn slowly changing representations, they are agnostic to the temporal structure of the visual experience, e.g., if toddlers look at an object for a long time before saccading to a different object.

## 3 METHOD

Our objective is to explore whether bio-inspired models of visual learning can utilize the actual eye-tracking derived visual experience of toddlers to develop robust object representations. To mimic toddlers' central visual experience, we use an eye-tracking dataset recorded during toddlers' play sessions and extract parts of frames centered on the gaze location. For comparison, we also simulate different visual experiences following alternative gaze strategies. Then, we train bio-inspired SSL models based on temporal slowness.

### 3.1 TODDLER FIXATION DATASET

The (Bambach et al., 2018) dataset contains head-camera videos recorded at 30 FPS and eye-tracking data for 38 dyads of toddlers/caregivers. All dyads play with the same set of 24 toys for 12 minutes on average. The children's ages range from 12.3 to 24.3 months. For 30 dyads, a head-camera resolution of $640 \times 480$ pixels was used, while four dyads were recorded at $720 \times 480$ pixels and the remaining four at $320 \times 240$ pixels. The horizontal field of view covers 72 degrees. Figure 1A shows an example video frame with the gaze location (Bambach et al., 2018). In the following, we explain how we simulate different gaze strategies by deriving several datasets from these play sessions. Additionally, we include the anonymized information of all toddlers who participated in the study in Appendix C.

**Toddler fixation dataset.** This dataset aims to simulate the central visual experience of toddlers. We cut out an image patch centered on the gaze point. For the cut out's size, we choose $128 \times 128$ pixel as the default, which corresponds to $14° \times 14°$ of visual angle. A typical temporal sequence of this dataset is illustrated in Figure 1B. If the gaze fixation point is too close to the image border, the crop boundaries may extend beyond the image, making it impossible to extract a patch of the desired size. In this case, we shift the gaze fixation point from the problematic border orthogonally by the minimum number of pixels. This ensures that the cropping operation outputs an image with the correct size. Note that the cropped area always contains the gaze fixation point. This dataset contains 559,522 training images, and this number is consistent across all fixation datasets (see below).

**Adult fixation dataset.** We want to investigate the differences between gaze fixation in adults and toddlers and the consequences of these differences on learned representations. Thus, we also extract image patches around adults' gaze fixation points following the procedure of the Toddler fixation dataset. Appendix A illustrates the gaze distributions of toddlers and adults.

**Random fixation dataset.** As a simple comparison dataset, we propose to simulate a completely random gaze strategy. We crop each frame around a location that is sampled uniformly at random. Unlike the Toddler/Adult fixation datasets, this dataset shows little spatio-temporal structure, and the cropped images are unlikely to contain well-centered objects. Figure 1D provides example frames from the Random fixation dataset.

**Centroid fixation dataset.** We also propose a stronger comparison dataset that considers a human moving their head but not their eyes. This is an important comparison because it distills the effect of eye gaze. One possibility could be to always crop the center of the frames. However, we noticed that the head-camera was often misaligned with respect to the stationary position of the eyes, resulting in a mismatch between the center of the frames and the center of the camera wearer's field of view (cf.

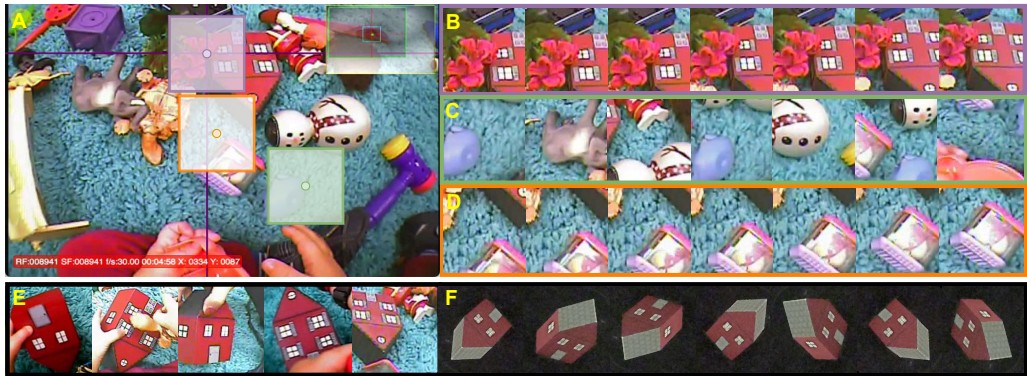

Figure 1: Examples of visual sequences for each of our datasets. **A.** Raw frame from an egocentric video with the locations of our different croppings. Purple, orange, and green boxes representing gaze fixation, centroid fixation, and random fixation, respectively. The cross indicates the gaze location given by the eye-tracker. **B-F.** Example sequences for **B-** the Toddler fixation dataset; **C-** the Random fixation dataset; **D-** the Centroid fixation dataset; **E** the Objects fixation dataset and **F-** the Plain background dataset. Note that datasets **E-F** have been manually curated to only contain views of the target objects. This kind of oracle knowledge is not available to a naive learner.

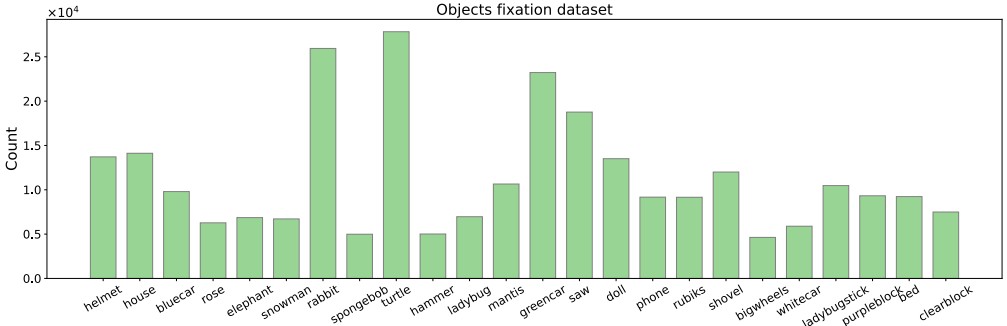

Figure 2: The number of images per object category in the Objects fixation dataset.

Appendix A). Thus, we rather use the centroid of the gaze fixation points (one for each participant video). To compute these centroids, we gather all gaze fixation points and calculate each video's mean of their horizontal and vertical coordinates. Note that, despite the centroid positions being fixed, the continuous movement of the head changes the visible portions of the scene. Nevertheless, compared to the Random fixation dataset, this set contains image patch sequences that are relatively stable over time. Figure 1D presents a temporal sequence of the Centroid fixation dataset.

We also consider "oracle" datasets that were constructed using the ground truth about an object's identity/location. Models trained on this dataset aim to upper-bound our model.

**Objects fixation dataset.** This dataset was collected from the same video frames used in the Toddler fixation dataset. Images were manually filtered such that toddlers looked at one of the target objects. From these frames crops with a 30-degree field of view around the gaze location were extracted, containing the target object while minimizing background interference (Bambach et al., 2018; Tsutsui et al., 2021). This dataset contains 271,754 images. Figure 1E displays examples of images. The number of images per toy is depicted in Figure 2, which indicates that the dataset is imbalanced. We conduct additional analysis on the class imbalance in Appendix B.3.

**Plain background dataset.** The Plain background dataset contains 128 viewpoints, capturing each object from various angles and distances for 1,536 images. Each image in this dataset displays a complete object against a black background, ensuring visual isolation from external distractions. Figure 1E shows an example toy from different viewpoints.

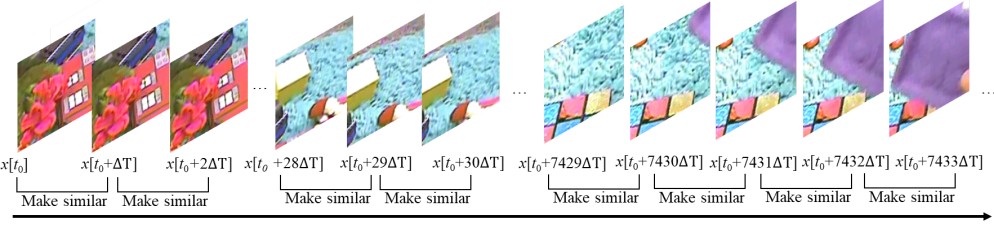

Figure 3: Illustration of SimCLR-TT on the Toddler fixation dataset. By default the time interval $\Delta T = \frac{1}{30}$ s corresponds to the inverse of the camera's frame rate, but it can be increased to an integer multiple of this value.

## 3.2 SELF-SUPERVISED LEARNING THROUGH TIME

To model the learning process of humans, we learn visual representations with a self-supervised model of temporal slowness, namely SimCLR-TT (Schneider et al., 2021). This algorithm is based on the state-of-the-art SimCLR method (Chen et al., 2020). SimCLR-TT samples an image $x_t$ at time $t$ and a temporally close image $x_{t+\Delta T}$ and computes their respective embeddings $z_t$, $z_{t+\Delta T}$ with a deep neural network (e.g. a ResNet). Unless stated otherwise, we set $\Delta T$ to the inverse of the camera's frame rate, i.e., $\Delta T = \frac{1}{30}$ seconds. In Section 4.3 we show additional results varying $\Delta T$. Then, SimCLR-TT minimizes

$$\mathcal{L}\left(z_t, z_{t+\Delta T}\right) = -\log \frac{\exp\left(\text{sim}\left(z_t, z_{t+\Delta T}\right)/\tau\right)}{\sum_{z_k \in \mathcal{B}, k \neq t}\left[\exp\left(\text{sim}\left(z_t, z_k\right)/\tau\right)\right]}, \tag{1}$$

where $\mathcal{B}$ is a minibatch, $\text{sim}(\cdot)$ is the cosine similarity and $\tau$ is the temperature hyper-parameter. Here $k \neq t$ but $k = t + \Delta T$ is possible. Thus, SimCLR-TT maximizes the similarity between temporally close representations (numerator) while keeping all representations dissimilar from each other (denominator). Figure 3 illustrates the learning process of SimCLR-TT. In Appendix B.1 we also present results for BYOL-TT (Schneider et al., 2021).

## 3.3 TRAINING AND EVALUATION

We run three random seeds for all experiments. For each random seed, we split the 38 available dyads into 30 train dyads and 8 test dyads. We train the models on train dyads for 100 epochs with a ResNet18, the AdamW optimizer, and set the initial learning rate and weight decay to $10^{-2}$ and $10^{-4}$, respectively. We set the SimCLR temperature to 0.08 and the batch size to 256. Appendix B.5 presents the results under various settings of hyper-parameters. We conduct all experiments on an Nvidia GeForce RTX 3090 GPU with 24 GB memory.

We assess the quality of the learned representations by training a linear classifier on top of the learned representation (right after the average pooling layer) in a supervised fashion (Chen et al., 2020). Since our pre-training datasets do not have labeled images, we always train the linear classifier on the train split of the Objects fixation dataset (same dyad's train split as for pre-training) and evaluate the object recognition accuracy on the test split of the Objects fixation dataset.

## 4 RESULTS

We aim to investigate whether toddlers' gaze behavior during a play session supports learning view-invariant object recognition. We also want to analyze the factors contributing to this.

### 4.1 TODDLERS' CENTRAL VISUAL FIELD EXPERIENCE SUPPORTS THE LEARNING OF INVARIANT OBJECT REPRESENTATIONS VIA TIME-BASED SSL

To test if a toddler's gaze behavior supports the learning of strong object representations, we compare the representations learned by SimCLR-TT when trained on the different datasets introduced in

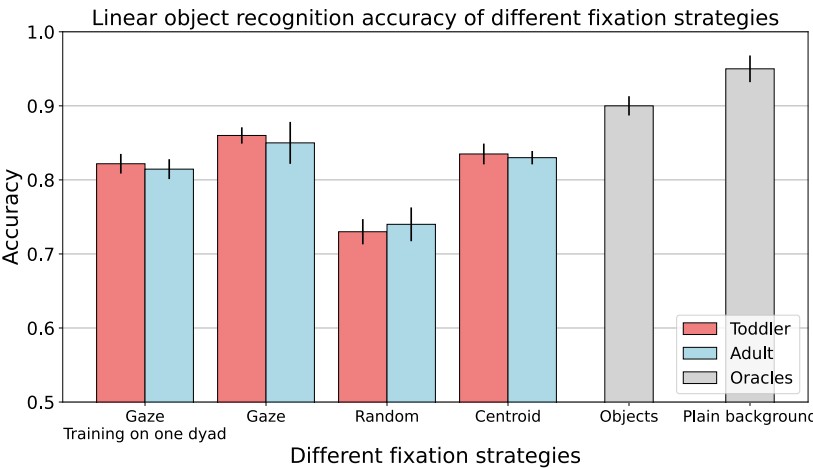

Figure 4: Linear object recognition accuracy of training on individual participants and all participants. We use red and blue to represent the test results of the model trained on the relevant datasets for toddlers and adults, respectively. The two gray bars represent the Oracles and indicate the test results on the Plain background and Object fixation datasets. Specifically, in the experiment, Oracles refers to the Objects fixation dataset and the Plain background dataset. For gaze fixation, we compared the test results of the model trained on each participant and across all participants. Specifically, the mean accuracy is shown, when trained with 38 dyads individually. For the all-participants setting, 30 participants are randomly selected from the pool of 38 for the training set under each random seed. The vertical bars represent the standard deviation over three random seeds.

Section 3.1 (Figure 4). Results for BYOL-TT (Schneider et al., 2021) show a similar trend and are given in Appendix B.1. We find that models trained with the Toddler fixation dataset outperform those trained with the Random fixation dataset (toddler) or the Centroid fixation dataset (toddler). This suggests that biologically inspired visual learning models like SimCLR-TT can leverage human gaze behavior to learn invariant object representations.

We wondered whether the visual experience of only **a single** human during a play session suffices to build good visual representations. To investigate this question, we train SimCLR-TT on the individual ecordings of each toddler and adult separately and compute the average of linear accuracies. We train the encoder (ResNet18) using all fixation data from a single toddler/adult, followed by training and testing the linear classifier with the Objects fixation data from the same and different toddlers/adults. We control the training set to comprise 75% of the total data, ensuring that the test set does not overlap with the training set. Figure 4 shows that the central visual experience of one toddler leads to representations almost as good as those from the central visual experiences of all toddlers. We show additional results with a larger ResNet50 in Appendix B.2.

Finally, we assess whether toddlers' visual experience produces better or worse representations than that of adults. By comparing the object recognition accuracy of models trained on fixation datasets from toddlers and adults, we see the same results. Toddlers' experiences induce more robust representations compared to adults when training with one toddler/adult trial, as well as when training with the entire dataset. We conclude that, toddlers' central visual experience supports more data-efficient learning than adults. Overall, toddlers appear to successfully curate their gaze behavior to permit the learning of robust object representations.

### 4.2 CONSTRAINING INPUT TO THE CENTRAL VISUAL FIELD IMPROVES LEARNING

Previous computational studies have neglected the importance of the constrained size of the central visual field for learning visual representations (Orhan et al., 2020; Sheybani et al., 2024). Here, we assess whether our simulated central visual experience leads to better/worse object representations than a wide field of view. We vary the crop size applied to the datasets reported in Section 3.1. In

Table 1: Linear object recognition accuracy for different cropping sizes. We have bolded the main results of gaze fixation, while the underlined results represent simulations that do not utilize actual gaze fixation and consider only the egocentric visual experience.

|  |  | $64 \times 64$ | $128 \times 128$ | $240 \times 240$ | $480 \times 480$ |
|---|---|---|---|---|---|
| Gaze fixation | Toddler | $0.831 \pm 0.015$ | $\mathbf{0.863 \pm 0.011}$ | $0.828 \pm 0.014$ | $0.805 \pm 0.018$ |
|  | Adult | $0.826 \pm 0.013$ | $\mathbf{0.851 \pm 0.028}$ | $0.816 \pm 0.013$ | $0.791 \pm 0.019$ |
| Random fixation | Toddler | $0.701 \pm 0.011$ | $0.736 \pm 0.017$ | $0.694 \pm 0.025$ | $0.589 \pm 0.036$ |
|  | Adult | $0.716 \pm 0.021$ | $0.742 \pm 0.022$ | $0.685 \pm 0.023$ | $0.576 \pm 0.019$ |
| Centroid fixation | Toddler | $0.822 \pm 0.016$ | $0.838 \pm 0.010$ | $0.815 \pm 0.018$ | $\underline{0.784} \pm \underline{0.022}$ |
|  | Adult | $0.818 \pm 0.012$ | $0.829 \pm 0.009$ | $0.807 \pm 0.014$ | $\underline{0.763} \pm \underline{0.017}$ |

Table 1, we observe for both toddlers and adults that an image size of $128 \times 128$ (corresponding to $14° \times 14°$ of visual angle) produces the best recognition accuracy for all gaze strategies. Importantly, Toddler and Adult gaze fixations $128 \times 128$ present an accuracy boost of $8\%$ compared to Centroid gaze $480 \times 480$, which simulates head-camera recordings without an eye-tracker. We conclude that accounting for the constrained size of the central visual field is crucial for learning powerful object representations. We speculate that, this boost originates in the ability of a $128 \times 128$ gaze-centered crop to frequently capture the complete structure of an object while minimizing irrelevant background information.

### 4.3 TODDLERS' GAZE BEHAVIOR FAVORS STRONGER EMPHASIS ON SLOWNESS

Previous work suggests that semantic aspects of the visual experience vary more slowly for toddlers than for adults (Sheybani et al., 2023) and that extending the gap of time between two positive pairs can improve the quality of object representations if visual inputs are sufficiently stable over time (Aubret et al., 2022; Schneider et al., 2021). Thus, we investigate whether amplifying the temporal slowness of our representation intensifies the difference between toddlers' and adults' representations. To amplify temporal slowness, we increase the temporal gap $\Delta T$ between representations that are made similar. As shown in Figure 5A, $\Delta T$ ranges from $\frac{1}{30}$ to 3.0 seconds, increasing continuously by 0.5 seconds at each step. The models trained with the Toddler fixation dataset achieve the highest recognition accuracy when $\Delta T = 1.5s$. Conversely, Figure 5B shows that, for models trained with the Adult fixation dataset, increasing the interval between positive pairs decreases the quality of object representations. The results are consistent for both human fixations and centroid fixations ("Fixation" and "Centroid"). We conclude that toddlers' gaze behavior favors a stronger emphasis on slowness (greater $\Delta T$) than that of adults.

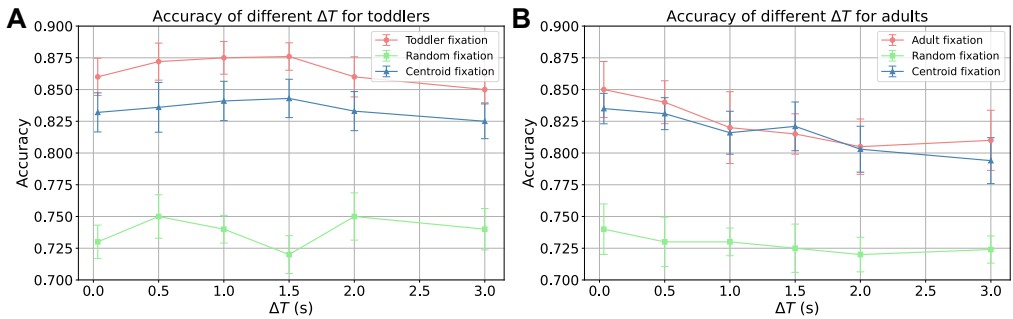

Figure 5: The impact of different $\Delta T$ on recognition accuracy for toddlers (**A**) and adults (**B**).

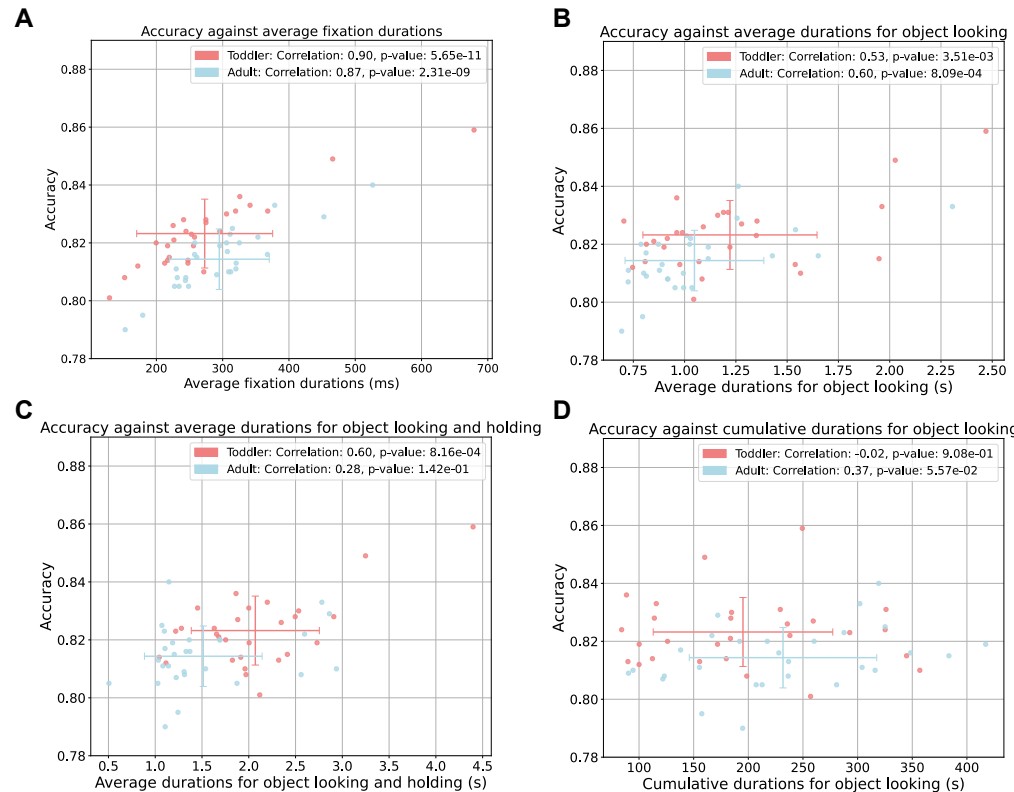

Figure 6: Correlation analysis between the linear recognition accuracy and the average fixation duration (**A**), the average duration of object looking (**B**), the average duration of object looking while holding the object (**C**), and the cumulative duration of object looking (**D**). Models were all trained on individual Toddler and Adult fixation datasets. In each figure, the crosshairs represent the mean and standard deviation of the data values over the two axes. The legends show the Pearson correlation coefficients and their p-values.

## 4.4 TODDLERS' LONG OBJECT INSPECTIONS RELATIVE TO ADULTS FACILITATE LEARNING

So far, we have shown that the egocentric visual experience of toddlers facilitates the self-supervised learning of object representations relative to that of adults. However, the temporal properties responsible for this effect remain unclear. Here, we further analyze the visual statistics of central visual experiences. We focus on four metrics that characterize the temporal sequence of images: the average fixation duration before making a saccade, the average duration of object looking bouts, the average duration of object looking when the camera-wearer holds the object, and the cumulative duration of object looking in a recording. We explain how we detect saccades and compute average fixation durations in Appendix A. For other metrics, we leverage manually labeled timestamps (by (Bambach et al., 2018)) about when toddlers and adults look at/hold an object. In the following, we label "Object looking" when the gaze fixation points are located on an object while the camera-wearer is not holding the object. We successfully extracted the data from 28 out of 38 toddlers and conducted all subsequent experiments using these 28 toddlers. The remaining participants are excluded from this section due to the lack of data on fixation durations. Table 3 in Appendix C presents the details of these specific 28 toddlers.

In Figure 6, we observe that object recognition accuracy is highly correlated with the three average durations but only weakly correlated with the cumulative duration of object looking. This indicates that long fixation bouts are important in explaining the relative quality of visual representations trained on the Toddler vs. Adult fixation datasets.

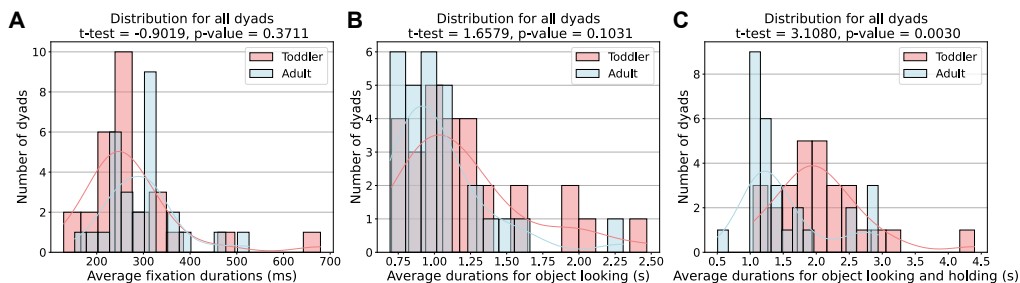

Figure 7: Comparison of average fixation duration **(A)**, average duration of object looking **(B)**, average duration of object looking while holding **(D)** for toddlers and adults. Each panel includes the frequency distribution for the given metric, along with a density curve. The t-test statistics and p-values are given in the titles.

The data in Figure 6 also allow us to confirm that on average toddlers' visual experience permits learning better representations than that of adults (t-test p-value $= 0.0053 < 0.05$), confirming our finding in Section 4.1 with the given subset of dyads. To investigate which metric plays a crucial role in the differences between toddlers and adults, Figure 7 presents the distributions of average fixation duration for toddlers and adults. We observe that toddlers look longer at the object that they are holding, in comparison with adults (t-test p-value $= 0.003 < 0.05$). Other metrics do not present statistically significant differences between adults and toddlers. We conclude that, compared to adults, toddlers' longer periods of object observation when manipulating the object allow learning better view-invariant object representations.

## 5 CONCLUSION

Current SSL approaches still struggle to learn robust human-like object representations and the reasons for this remain unclear. Here, we investigated whether biologically inspired visual learning models can take advantage of toddlers' gaze behavior to develop robust object representations. We cropped the toddlers' gaze location from egocentric video recordings with eye-tracking during play sessions. Then, we trained bio-inspired unsupervised models that drive visual representations to slowly change. Our findings indicate that toddlers' gaze strategies permit the learning of representations that support view-invariant object instance recognition within a single play session of 12 minutes. Results were weaker for adults' gaze behavior. Our analysis shows that our approximated central visual experience is crucial for learning object-oriented representations and that toddlers' gaze behavior favors a stronger emphasis on slowness compared to adults. This is consistent with toddlers looking longer at objects while holding them. During their relatively long holding periods, toddlers may turn and move the object, giving access to high-quality sequences containing different object views over a short period of time.

From a developmental perspective, our work provides strong evidence that the development of view-invariant representations can originate from a slowness learning objective, a mechanism supported by neuroscientifc studies (Li & DiCarlo, 2008; Miyashita, 1988). We further demonstrate that toddlers may curate their gaze behavior to enhance the quality of their visual representations. From a machine learning perspective, we show that combining eye-tracking video data and SSL supports unsupervised view-invariant recognition. This work marks a significant step towards learning strong representations without hand-crafted image datasets (e.g., (Aubret et al., 2022)).

We analyzed gaze behavior in toddlers with a minimum age of 12.3 months, meaning they had substantial visual learning experience before the experiment, while our models learned from scratch. Expanding to a wider variety of objects and participants, particularly younger toddlers with distinct visual exploration patterns, could offer deeper insights into early visual representation development. Studying how babies under one year engage with objects may reveal new aspects of gaze behavior that contribute to visual learning (Maurer, 2017; Sheybani et al., 2024). Moreover, refining our approach to incorporate both central and peripheral vision could provide a more accurate simulation of human perception (Wang et al., 2021).

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

# A    ADDITIONAL DETAILS

**Gaze location distribution.**    In section 3.1, we explain that the center of the frames is misaligned with respect to the stationary position of the eyes. To support this statement, Figure 8 and Figure 9 display the distribution of gaze locations for each toddler and adult, respectively. Brighter areas indicate higher frequencies of gaze fixation at those locations. The results indicate that their average gaze location is not centered with respect to the camera.

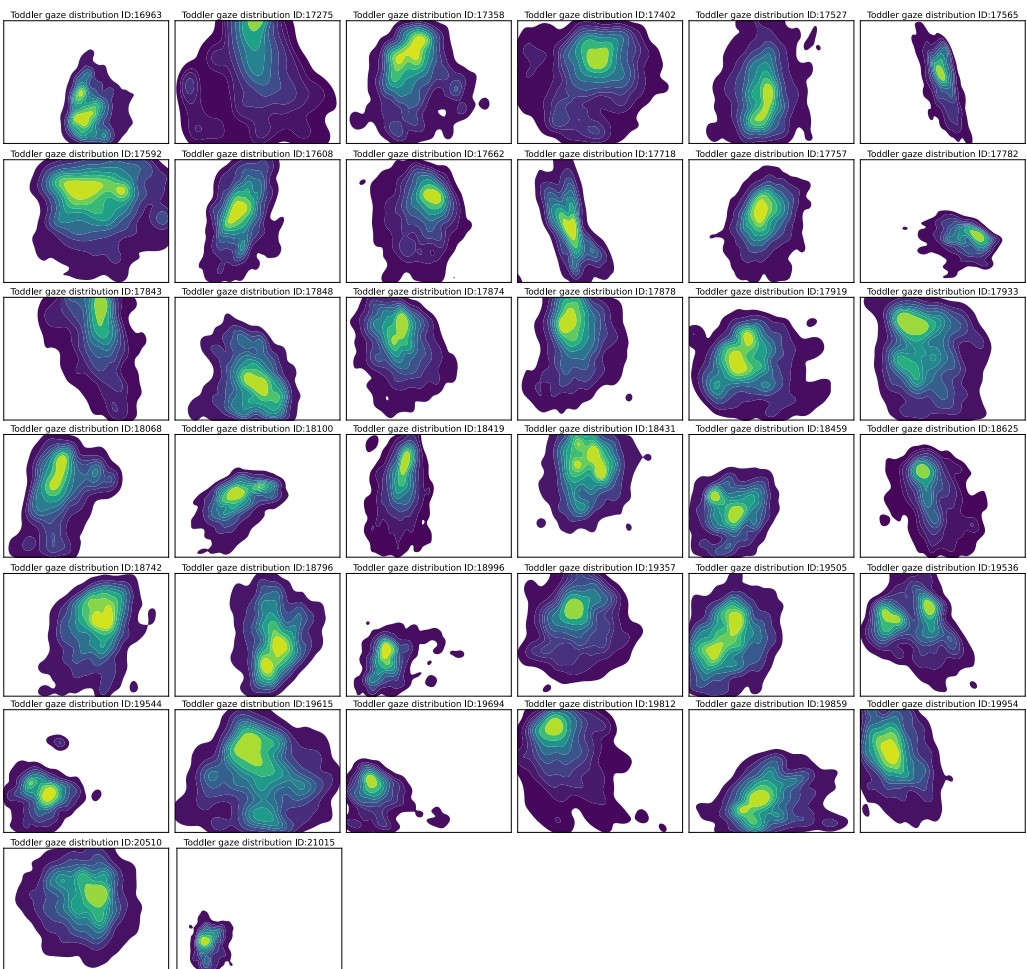

Figure 8: Gaze distribution for all toddlers.

**Extraction of saccade and fixations.**    In the study in section 4.4, we extracted fixation bouts. This requires to detect saccades, as they bound the fixations bouts. To detect saccades in gaze movement, we apply a velocity threshold-based method similar to (Raabe et al., 2023). Consecutive gaze points that exceed a threshold $T_1$ are identified as a single saccade. To account for artifacts caused by low frame rates, a second threshold $T_2$, along with an angular criterion $\theta$, allows the inclusion of the two data points adjacent to the saccade initially detected. Any data points not classified as saccades are considered fixations. For this study, we choose $T_1 = 25\,^\circ\,\mathrm{s}^{-1}$, $T_2 = 10\,^\circ\,\mathrm{s}^{-1}$ and $\theta = 45^\circ$.

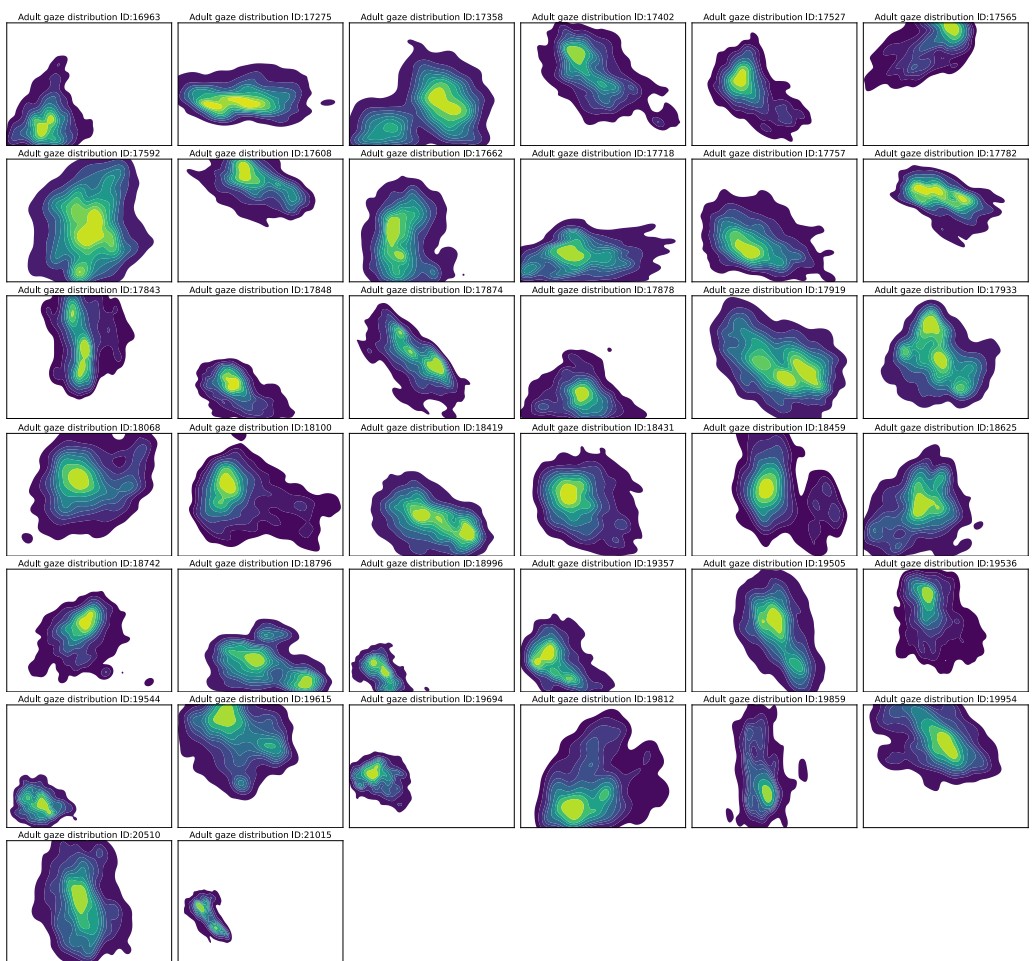

Figure 9: Gaze distribution for all adults.

# B COMPLEMENTARY ANALYSIS

## B.1 RESULTS OF TRAINING BYOL-TT

In order to evaluate whether our conclusions also hold for different methods learning with temporal slowness, we perform the same experiments described in Section 4.1 with BYOL-TT. Similar to SimCLR-TT, BYOL-TT was originally considered to be used for contrastive learning through time (Schneider et al., 2021). Its loss function is defined as

$$\mathcal{L}_{\theta_t, \xi_{t+\Delta T}} = 2 - 2 \cdot \text{sim}\left(q_{\theta_t}\left(z_{\theta_t}\right), z_{\xi_{t+\Delta T}}\right), \tag{2}$$

where $q_{\theta_t}(z_{\theta_t})$ is the prediction of the online network for one frame, $z_{\xi_{t+\Delta T}}$ represents outputs from the target network. Here, $\theta$ corresponds to the weights of the online network, and $\xi$ represents the weights of the target network. Again, we use the cosine similarity as the similarity function.

In Figure 10A, we found, in line with (Schneider et al., 2021) that BYOL-TT, as the backbone model, extracts less effective representations from the different fixation strategy datasets compared to SimCLR-TT. However, the relative relationships between the data remain unchanged. Overall, the conclusion that toddler fixation contributes to the acquisition of more robust representations still holds.

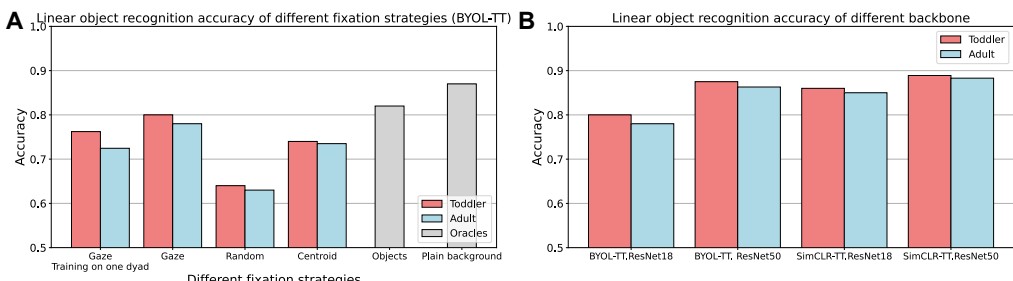

Figure 10: Linear object recognition accuracy of different settings. **A.** Testing results of BYOL-TT training on different datasets. **B.** Different backbone training on Toddler and Adult fixation dataset.

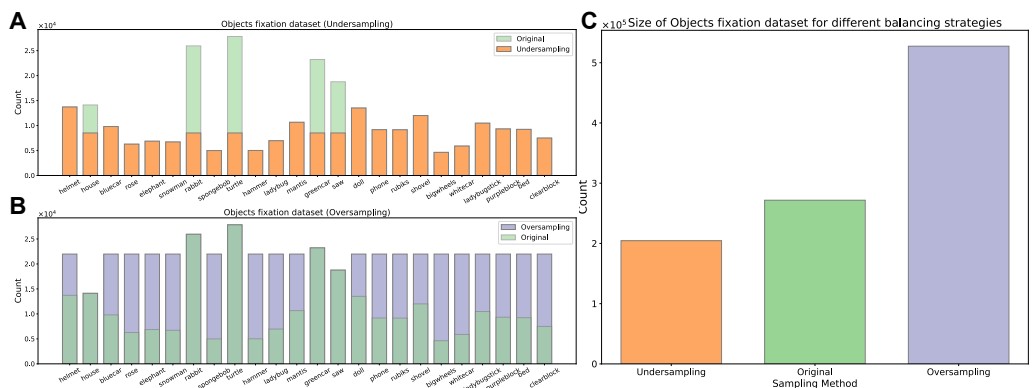

Figure 11: Sampling on Objects fixation dataset. **A.** Undersampling; **B.** Oversampling; **C.** Comparison of dataset sizes after using different sampling methods.

## B.2 IMPACT OF CHANGING THE SELF-SUPERVISED LEARNING ENCODER

We compared the accuracy of BYOL-TT and SimCLR-TT using ResNet-18 and ResNet-50 as encoders on both the Toddler and Adult fixation datasets. As shown in Figure 10B, introducing more complex encoders resulted in a significant improvement in accuracy, with the gap between toddler and adult performance narrowing. This suggests that a more sophisticated encoder can equalize different boosting sampling strategies, which may obscure the inherent differences in representations between toddlers and adults. In contrast, a simpler encoder tends to profit more from toddler gaze behavior compared to those from adults.

## B.3 ANALYSIS OF THE CLASS IMBALANCE IN THE OBJECTS FIXATION DATASET

To investigate the impact of the imbalance in the Objects fixation dataset shown in Figure 2, we adjusted the distribution of the Objects fixation dataset while keeping the original encoder training results unchanged. The linear classifier was then trained and tested on the adjusted datasets. The number of categories remained fixed at 24 throughout the experiments. We compared the results of two types of sampling strategies:

**Undersampling.** We applied random undersampling to reduce the number of samples in the top 5 categories, making their quantities similar to those of the other categories. We do not intend to equalize all classes. In real-world scenarios, toddlers naturally show preferences for certain toys, and this behavior should be preserved. Our goal is to smooth the occurrence probabilities of other objects relatively rather than enforce an artificial balance across all categories.

**Oversampling.** Similarly, we applied random oversampling to increase the number of samples in the underrepresented categories to match the quantity of the top 5 categories. However, this method will result in duplicate samples in the dataset.

The data distributions after applying both sampling methods are shown in Figure 11. We maintain the experimental setup consistent with Section 4.1 and train a linear classifier on the undersampling and oversampling object fixation datasets.

In Figure 12, we observe that when the total sample size is reduced, the recognition accuracy of the models trained on Toddler and Adult fixation datasets decreases, but the difference in their accuracy continues to widen. However, with more complex or balanced training, the model's generalization capacity improves, and the performance across toddlers and adults tends to converge, reducing the impact of differences in visual behaviors. Therefore, toddler gaze behavior might offer a greater advantage under undersampling conditions.

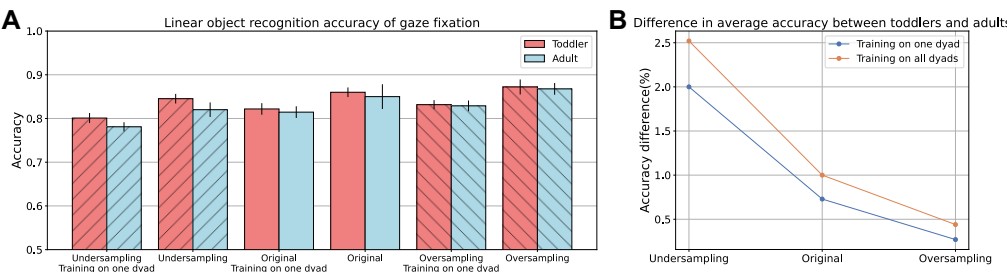

Figure 12: Linear object recognition accuracy and the difference in accuracy between undersampling and oversampling. **A.** We compared the recognition accuracy under different sampling methods, where "/" represents undersampling and "\" represents oversampling. Additionally, we provide the test results after training on one dyad versus all dyads; **B.** The difference in recognition accuracy between toddlers and adults under different sampling methods. Here, we also compare the accuracy differences of the model trained on one dyad versus all dyads.

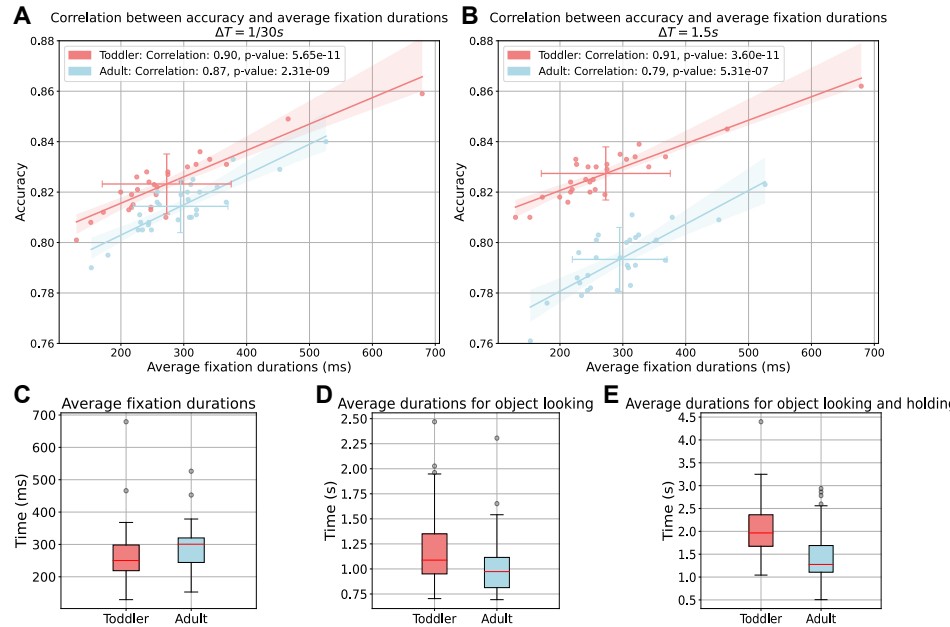

Figure 13: Some evidence highlights the differences between toddlers and adults. In **(A-B)**, We observe variations in the test accuracy of models training on the Toddler and Adult fixation datasets under different $\Delta T$. We attached fitted regression lines, and the shaded areas show the 95% confidence interval. **(C-E)** illustrates box plots showing the data differences between toddlers and adults across three metrics. The red line indicates the median value (Q2), and the gray dots represent outlier data exceeding the upper quartile (Q3).

### B.4 HIGHLIGHTS DIFFERENCES BETWEEN TODDLERS AND ADULTS

We provide additional evidence highlighting the differences between toddlers and adult. In Figure 13A and Figure 13B, we compare the changes in recognition accuracy for both toddlers and adults under different $\Delta T$ values. From the regression lines, the increasing $\Delta T$ amplifies the difference in recognition accuracy between training on toddlers' and adults' fixation datasets, consistent with the findings in Section 4.3. Besides Figure 7, Figure 13C-E display the box plots for the three corresponding metrics, revealing significant distinctions in the way toddlers and adults observe objects across all three metrics.

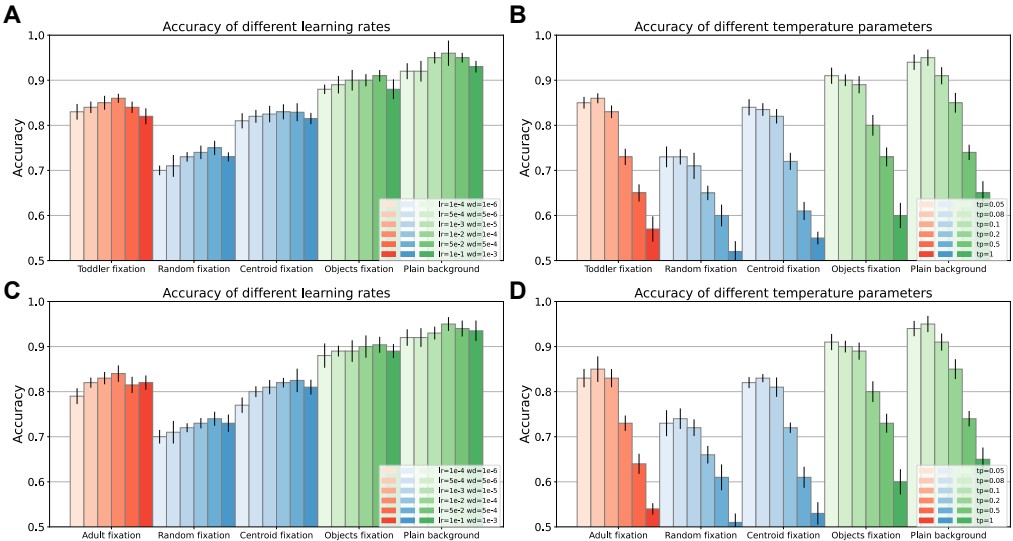

Figure 14: Object recognition accuracy across different hyper-parameter settings.

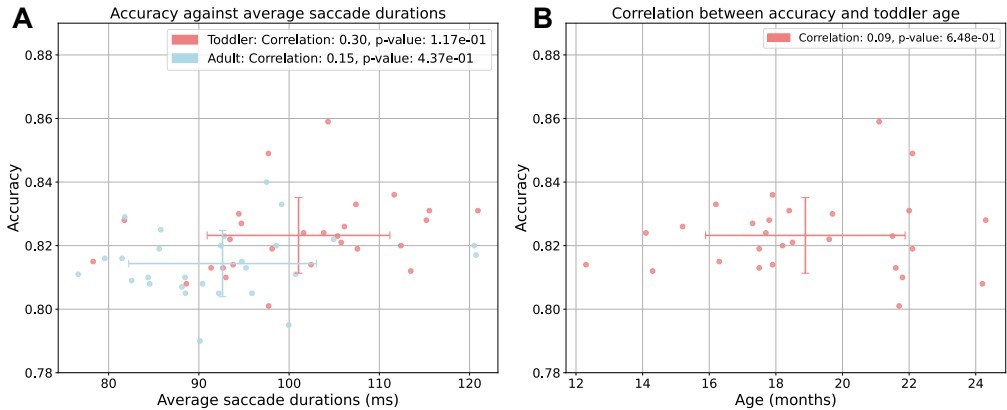

Figure 15: The impact of **(A)** average saccade duration and **(B)** toddler's age on recognition accuracy.

### B.5 ROBUSTNESS TESTING WITH VARYING HYPER-PARAMETERS

The learning rate (lr), weight decay (wd), and temperature (tp) used in our main content were selected as the best settings after fine-tuning. To assess the robustness of our method, we conducted additional experiments where we fixed the $lr = 10^{-2}$ and $wd = 10^{-4}$ and $tp = 0.08$ to and varying another hyper-parameter individually. As shown in Figure 14, changes in these hyper-parameters do not affect the conclusions presented in Section 4.1.

### B.6 STUDY OF SACCADE DURATION AND AGE

Here, we complement section 4.4 and study two additional metrics that may impact the performance of individual adults and toddlers, namely the average saccade duration and toddlers' age. According to Figure 15, we observe no significant correlation between the recognition accuracy and both the average saccade duration or toddlers' age. However, the youngest toddlers in the study were older than one year and we can not rule out that babies may induce different results.

## C DETAILS OF ALL TOYS AND TODDLERS DATA

We provide information for all toys and toddlers participating in the study in Table 2 and Table 3. The toddler ID represents an anonymized identifier for each toddler.

Table 2: 24 toys were used for toddler interaction. Among them, " Library " refers to those toys that were successfully recognized when the toddler calls any word from the corresponding row. However, these columns are not within the scope of the current study's discussion. The main focus is on the colors, shapes, or textures of these 24 toys, which are more likely to help toddlers differentiate between them.

| Gazetag Naming | ICONS | ID | Library |
|---|---|---|---|
| helmet | | 1 | helmet, hat |
| house | | 2 | house, home |
| bluecar | | 3 | car |
| rose | | 4 | rose, flower, plant |
| elephant | | 5 | elephant |
| snowman | | 6 | snowman |
| rabbit | | 7 | rabbit, bunny |
| spongebob | | 8 | spongebob, block |
| turtle | | 9 | turtle, tortoise |
| hammer | | 10 | hammer, tool, mallet |
| ladybug | | 11 | bug, insect, ladybug, beetle |
| mantis | | 12 | bug, insect, praying mantis, mantis, grasshopper |
| greencar | | 13 | car |
| saw | | 14 | saw, tool |
| doll | | 15 | baby, baby doll, girl, doll |
| phone | | 16 | phone, telephone |
| rubiks | | 17 | block, rubiks cube, rubiks, cube |
| shovel | | 18 | rake, shovel, tool |
| bigwheels | | 19 | truck, jeep, bigwheel, car |
| whitecar | | 20 | car, policecar |
| ladybugstick | | 21 | ladybug, bug |
| purpleblock | | 22 | block, cube |
| bed | | 23 | bed |
| clearblock | | 24 | block, cube |

Table 3: Information for each toddler participating in the study (Anonymized). The Toddler IDs marked with " * " indicate participants in the experiment in Section 4.4, while the Frame Count refers to the total number of video frames used in the dataset. Video Length specifies the recorded time interval of the video. Age refers to the toddler's age at the time of participation in the study. In the Gender column, M denotes male, and F denotes female. The Resolution specifies the recording resolution of the video recorded by the head-mounted camera.

| Toddler ID | Frame Count | Video Length | Age (months) | Gender | Resolution |
|---|---|---|---|---|---|
| 16963 | 16440 | 9:07 | 20.7 | M | 720x480 |
| 17275 | 9120 | 5:04 | 18.2 | F | 720x480 |
| 17358 | 18930 | 10:31 | 18.8 | M | 720x480 |
| 17402 | 27636 | 15:21 | 19.2 | M | 640x480 |
| 17527* | 15242 | 8:28 | 21.5 | M | 640x480 |
| 17565* | 14864 | 8:15 | 19.7 | F | 640x480 |
| 17592* | 16116 | 8:58 | 18.2 | M | 640x480 |
| 17608* | 18059 | 10:02 | 21.8 | F | 640x480 |
| 17662* | 14553 | 8:05 | 15.2 | F | 640x480 |
| 17718 | 11850 | 6:35 | 18.1 | F | 720x480 |
| 17757* | 19661 | 10:55 | 21.7 | F | 640x480 |
| 17782* | 9035 | 5:01 | 22.1 | F | 640x480 |
| 17843* | 18209 | 10:07 | 19.6 | F | 640x480 |
| 17848* | 21111 | 11:43 | 18.4 | F | 640x480 |
| 17874* | 17429 | 9:41 | 17.8 | M | 640x480 |
| 17878* | 20018 | 11:08 | 17.5 | F | 640x480 |
| 17919* | 18596 | 10:20 | 22.1 | M | 640x480 |
| 17933* | 14457 | 8:02 | 17.9 | F | 640x480 |
| 18068* | 7976 | 4:26 | 17.9 | M | 640x480 |
| 18100* | 14982 | 8:19 | 16.3 | F | 640x480 |
| 18419* | 28253 | 15:41 | 17.3 | M | 640x480 |
| 18431* | 11575 | 6:26 | 22 | M | 640x480 |
| 18459* | 7231 | 4:01 | 16.2 | F | 640x480 |
| 18625* | 18209 | 10:07 | 24.3 | F | 640x480 |
| 18742* | 19018 | 10:34 | 17.7 | M | 640x480 |
| 18796* | 11672 | 6:30 | 24.2 | M | 640x480 |
| 18996 | 12466 | 6:56 | 15.9 | F | 320x240 |
| 19357* | 8834 | 4:54 | 17.5 | M | 640x480 |
| 19505* | 18397 | 10:13 | 18.5 | M | 640x480 |
| 19536* | 18370 | 10:13 | 21.1 | M | 640x480 |
| 19544 | 9151 | 5:05 | 13.8 | F | 320x240 |
| 19615* | 13351 | 7:25 | 14.1 | M | 640x480 |
| 19694 | 10801 | 6:00 | 15.2 | M | 320x240 |
| 19812* | 9918 | 5:31 | 21.6 | M | 640x480 |
| 19859 | 7360 | 4:05 | 14.4 | M | 640x480 |
| 19954* | 9201 | 5:07 | 12.3 | F | 640x480 |
| 20510* | 11865 | 6:35 | 14.35 | M | 640x480 |
| 21015 | 9566 | 5:19 | 13 | M | 320x240 |

