# OpenReview forum: "Active Gaze Behavior Boosts Self-Supervised Object Learning"
_ICLR.cc/2025/Conference — ICLR 2025 Conference Withdrawn Submission_

### Official Review · Reviewer_BEUM · 2024-10-23

**Soundness:** 2
**Presentation:** 3
**Contribution:** 2
**Rating:** 3
**Confidence:** 4

**Summary:**

This paper explores whether a bio inspired visual learning model can harness toddlers’ gaze behavior during a play session to develop view-invariant object recognition. Exploiting head-mounted eye tracking during dyadic play, the authors simulate toddlers’ central visual field experience by cropping image regions centered on the gaze location. This visual stream feeds a time-based self-supervised learning algorithm. The experiments demonstrate that toddlers’ gaze strategy supports the learning of invariant object representations.

**Strengths:**

1. The idea of combining toddlers’ gaze behavior with self-supervised learning of view-invariant object recognition is interesting.
2. The writing style of this paper is clear and fluent, and the content is easy to understand.
3. The experimental part is quite detailed.

**Weaknesses:**

This paper claims to "reveal how toddlers’ gaze behavior supports self-supervised learning of view-invariant object recognition". However,
1. in Sec. 3.1, the proposed random/centroid fixation datasets, i.e, the control groups in experiments, do not contain objects of a specific category, making it difficult to prove the effectiveness of the proposed toddler/adult fixation datasets in the view-invariant object recognition task;
2. the entire work only validated on one self-supervised model, which is clearly lacking in persuasiveness;
3. in Sec. 4.1, the experimental results in Figure 4 do not support the authors' claim that "toddlers appear to curate their gaze behavior to develop robust object representations quickly". Firstly, there is no significant difference in the experimental results between adults and toddlers. Secondly, the results of toddler/adult fixation datasets are significantly inferior to the objects fixation dataset, indicating that toddler/adult fixation datasets cannot help the self-supervised model learn better representations.

**Questions:**

`Toddler' refers to children between the ages of 1 and 3. However, all the `toddler' participating in the experiment in Table 2 are between the ages of 12 and 25. Is it the true experimental setting, or is it a typographical error by the authors?

---

> ### Author Response · Authors · 2024-11-25
> **Response to Reviewer BEUM**
>
> Thank you for the careful reading and constructive feedback. We address each point raised below.
>
> - **W1: In Sec. 3.1, the proposed random/centroid fixation datasets, i.e, the control groups in experiments, do not contain objects of a specific category, making it difficult to prove the effectiveness of the proposed toddler/adult fixation datasets in the view-invariant object recognition task.**
>
> We think there is one or the other misunderstanding. Except for the two ''oracle'' datasets, none of the datasets are constructed to display objects in the selected image portion. Our objective is studying the importance of toddlers' gaze behaviors for learning object representations with temporal slowness. For this it is meaningful to compare toddler's actual gaze with simulated random eye movements or centroid fixation simulating no eye movements (but only head movement).
>
> - **W2: The entire work only validated on one self-supervised model, which is clearly lacking in persuasiveness.**
>
> We added experiments with another model, BYOL-TT in Appendix B.1, and also with larger networks in Appendix B.2. The results confirm our previous findings.
>
> - **W3: In Sec. 4.1, the experimental results in Figure 4 do not support the authors' claim that "toddlers appear to curate their gaze behavior to develop robust object representations quickly". Firstly, there is no significant difference in the experimental results between adults and toddlers. Secondly, the results of toddler/adult fixation datasets are significantly inferior to the objects fixation dataset, indicating that toddler/adult fixation datasets cannot help the self-supervised model learn better representations.**
>
> We would like to clarify that the object fixation dataset is an oracle dataset that can only be used as an upper-bound of the performance of our model. Toddlers do not have access to such oracle information telling them where in the scene all the relevant objects are.
>
> The differences between toddlers and adults are indeed small (albeit consistent across network models). We removed ''quickly'' from our claim to avoid the impression that toddler's visual input permits faster learning. However, we have highlighted other interesting differences between toddlers and adults in Appendix B.4.
>
> Overall, we think that our claim that toddlers’ gaze behavior supports self-supervised learning of view-invariant object recognition is well supported by our analyses.
>
> - **Q1: Toddler' refers to children between the ages of 1 and 3. However, all the toddler' participating in the experiment in Table 2 are between the ages of 12 and 25. Is it the true experimental setting, or is it a typographical error by the authors?**
>
> The age of considered toddlers ranges between 12.3 and 24.3 months, which aligns with the definition of toddlers. We have updated the ''Age'' column in the original Table 3 to ''Age (months)'' to avoid confusion.

---

### Official Review · Reviewer_6nrm · 2024-10-26

**Soundness:** 2
**Presentation:** 3
**Contribution:** 2
**Rating:** 5
**Confidence:** 5

**Summary:**

This workexamines how toddlers’ gaze helps self-supervised models learn view-invariant object recognition. Using eye-tracking data from toddler play sessions, the authors simulate a limited central vision experience and apply a time-based learning algorithm, SimCLR-TT. Results show that toddlers’ gaze patterns enhance object recognition better than random or adult gaze. Key contributions of this work include insights into how toddlers’ gaze supports learning and how such strategies could benefit ML models.

**Strengths:**

Originality: The paper introduces a unique, biologically inspired approach to SSL by modeling toddler gaze behavior to improve view-invariant object recognition. Rather than focusing on synthetic or curated datasets, the authors use eye-tracking data to simulate central vision dynamics in real-life toddler play sessions. This approach is a creative blend of insights from developmental psychology and machine learning, making it an original contribution.

Quality: The paper presents a thorough experimental design, comparing toddler and adult gaze strategies and establishing baselines (e.g., random and centroid fixation). The choice of the SimCLR-TT model, combined with a carefully curated dataset, shows a clear effort to test hypotheses in a controlled yet realistic setting. The analysis also includes multiple measures to validate that toddler gaze uniquely supports learning. These methodological choices enhance the paper’s quality and reinforce its claims.

Clarity: The paper is generally clear, with a well-structured explanation of the motivations behind gaze-centered learning and the potential benefits for SSL models. Concepts are explained effectively, with sufficient background to understand why toddler gaze was chosen as the focus.

Significance: This work has significant implications for both ML and developmental science. The study also contributes to understanding human learning processes, suggesting that toddler gaze behavior naturally supports efficient learning. This could inspire further interdisciplinary work and innovations in adaptive vision systems.

**Weaknesses:**

- The model is tested on a controlled dataset with limited objects, which might not generalize to real-world, diverse scenes. Expanding to more varied data or testing on real-world videos would make the findings more robust.

- Comparing toddler gaze data only with adult and random gaze is not sufficient. Adding standard data augmentations like rotation or scaling would better show if gaze-based learning offers unique advantages.

- The model only uses central vision, though peripheral vision plays a big role in human perception

- The paper lacks insight into which features are affected by toddler gaze.

- Evaluating only on object recognition limits the findings.

**Questions:**

- Why focus only on central vision? How about peripheral vision’s role in learning?
- How exactly does gaze duration affect feature learning? Any patterns in which features are learned?
- Have you addressed overfitting?
- Any ideas how to evaluate beyond object recognition, like for object localization?
- The dataset is unbalanced. Did you do analysis on that?
- Are toddler vs. adult differences statistically significant? Have you tested with random age samples?
- Are there any visible differences in feature maps from toddler and adult data? Seen patterns in views?
- Do you have evidence that toddlers learn with temporal slowness, or is this an assumption?

---

> ### Author Response · Authors · 2024-11-25
> **Response to Reviewer 6nrm [1/2]**
>
> We sincerely thank you for reading our manuscript and for your detailed questions and comments. Our responses are as follows.
>
> - **W1: The model is tested on a controlled dataset with limited objects, which might not generalize to real-world, diverse scenes. Expanding to more varied data or testing on real-world videos would make the findings more robust.**
>
> The dataset used in this study consists of video recordings capturing real-world interactions between toddlers and toys, with accurate gaze points tracked during these interactions. Although the controlled environment included only 24 different object categories, which limits the dataset's diversity, our primary aim is to explore the impact of toddlers' gaze patterns on learning representations. Given that the data is derived from authentic video recordings, we contend that this dataset is sufficient to support our research claims.
>
> - **W2: Comparing toddler gaze data only with adult and random gaze is not sufficient. Adding standard data augmentations like rotation or scaling would better show if gaze-based learning offers unique advantages.**
>
> Thank you for your suggestion. Our current focus is on the cognitive processes involved in toddlers' learning. Traditional data augmentation techniques, such as image rotation or scaling, cannot be considered biologically plausible. There is no evidence for such operations in the visual system.
>
> However, we acknowledge that comparing these methods from a machine-learning perspective is intriguing. In future work, we aim to thoroughly investigate the interplay between classical image augmentations and natural gaze behavior to deepen our understanding of their respective benefits in learning representations.
>
> - **W3: The model only uses central vision, though peripheral vision plays a big role in human perception.**
>
> We acknowledge the importance of peripheral vision in perception. Our study emphasizes the central visual field because it has high acuity and is critical for tasks that require detailed visual processing, such as object recognition. Our experiments with different crop sizes show a tangible benefit of ignoring the periphery when learning invariant representations of objects that need to be distinguished in the same context. In other situations, a wider context may help to identify the solution in a challenging recognition task.
>
> - **W4: The paper lacks insight into which features are affected by toddler gaze.**
>
> We acknowledge that the current paper does not explicitly detail which specific features are influenced by toddler gaze patterns. Our primary focus has been on establishing the broader relationship between gaze behavior and learning object representations.
>
> - **W5: Evaluating only on object recognition limits the findings.**
>
> Our main contribution is showing that toddlers' gaze behavior supports the learning of invariant object representations. We believe our study provides sufficient evidence to support this claim. Toddlers presumably learn many other things during such interactions, but these are simply outside of the scope of the current study.

---

> ### Author Response · Authors · 2024-11-25
> **Response to Reviewer 6nrm [2/2]**
>
> - **Q1: Why focus only on central vision? How about peripheral vision’s role in learning?**
>
> Our study focuses on the central visual field because it is critical for object recognition[1-2]. We leave to future works the study of peripheral vision, including how it can be modeled and how it further impacts object recognition.
>
> [1] Quaia, Christian, and Richard J. Krauzlis. "Object recognition in primates: what can early visual areas contribute?." Frontiers in Behavioral Neuroscience 18 (2024): 1425496.
>
> [2] Yu, H-H., T. A. Chaplin, and M. G. P. Rosa. "Representation of central and peripheral vision in the primate cerebral cortex: Insights from studies of the marmoset brain." Neuroscience Research 93 (2015): 47-61.
>
> - **Q2: How exactly does gaze duration affect feature learning? Any patterns in which features are learned?**
>
> We're unsure if we understand the question. In general, for learning of invariant representations via a slowness objective to work, successive views should show the same object from different perspectives. This can be achieved by gaze being still while the object moves or the object being still while gaze moves or a combination of the two. At any rate, it is important that gaze is likely to rest on the same object from one moment to the next, see, e.g., [3].
>
> [3] Schneider, Felix, et al. "Contrastive learning through time." SVRHM 2021 Workshop@ NeurIPS. 2021.
>
> - **Q3: Have you addressed overfitting?**
>
> Yes, we use a lightweight encoder (ResNet18) with weight decay and batch normalization during training. We split the train and validation set and do not see a decrease in validation accuracy.
>
> - **Q4: Any ideas how to evaluate beyond object recognition, like for object localization?**
>
> For now we do not have a clear idea of how to evaluate object localization with the same set of objects (to avoid a distribution shift). We are open to suggestions.
>
> - **Q5: The dataset is unbalanced. Did you do analysis on that?**
>
> We now add additional analyses to evaluate the impact of dataset imbalance. Please see Appendix B.3 for details. In summary, toddler gaze behavior may provide a greater advantage under undersampling conditions.
>
> - **Q6: Are toddler vs. adult differences statistically significant? Have you tested with random age samples?**
>
> We show in Section 4.4 that the difference is statistically significant. Could you please clarify what the mean by ''random age samples''? The subject data collection procedures are described in [4] (Section 3.1).
>
> [4] Bambach, Sven, et al. "Toddler-inspired visual object learning." Advances in neural information processing systems 31 (2018).
>
> - **Q7: Are there any visible differences in feature maps from toddler and adult data? Seen patterns in views?**
>
> We have not yet explored the comparison of their feature map patterns. Currently, we focus on the recognition accuracy of toddlers and adults, aiming to intuitively reflect the differences in feature extraction by the model through their respective accuracies. Additionally, we have further compared the differences in the way toddlers and adults observe objects.
>
> - **Q8: Do you have evidence that toddlers learn with temporal slowness, or is this an assumption?**
>
> It's currently the best supported theory. Please see Section 2 (Computational studies of visual learning with temporal slowness). In addition, [5] specifically highlights the relationship between toddlers and temporal slowness in Section 6.3.
>
> [5] Sheybani, Saber, et al. "Curriculum learning with infant egocentric videos." Advances in Neural Information Processing Systems 36 (2024).

---

> > ### Comment · Reviewer_6nrm · 2024-11-26
> > **Comments on Rebuttal**
> >
> > Thank you for your answers and clarifications.

---

> ### Author Response · Authors · 2024-11-26
> **Response by Authors**
>
> Thank you for your feedback. We have made an additional effort to improve the manuscript’s clarity regarding our aims, contributions, and conclusions. Please review the updated manuscript along with the official comments. We would appreciate any additional comments or questions you may have.

---

### Official Review · Reviewer_74Gv · 2024-11-02

**Soundness:** 2
**Presentation:** 2
**Contribution:** 2
**Rating:** 6
**Confidence:** 3

**Summary:**

The paper investigates the adoption of a bio-inspired model of visual learning based on the visual experience of toddlers (compared to adults) captured thanks to the use of an eye-tracker. The hypothesis is that data collected from toddlers from a first-person perspective can improve the robustness of object representations, in particular in the presence of data collected from multiple views.
The authors assess their method on a dataset of head-camera recordings and gaze tracking from toddlers and adults during play sessions.

**Strengths:**

- The paper considers an inspiring scientific question, at the crossroads between different disciplines, and as such it may be of interest for a multi-disciplinary community
- The work is well-motivated and contextualised concerning the existing literature

**Weaknesses:**

- Besides the scientific question, which I find very interesting indeed, it's not clear to me what would be the use and the impact of this analysis. What are the practical scenarios where these results can be exploited? How the empirical observations that you are making can be turned into actionable insights? For instance, can the insights be turned into suggestions on the way datasets for object recognition should be collected?
- The authors gather strong conclusions from their experiments. However, in my opinion, the provided experiments are not enough to speak in favour of the generality of the outcomes (see Questions)

**Questions:**

- The contributions should be clearly stated in the introduction, to fully unveil the intentions of the authors. For instance, are the datasets a contribution? Will you share the data with the community?

- About the datasets: (1) Can you please explain more clearly the motivations behind the centroid fixation dataset? How can you be sure that an object is always present in the selected image portion? (2) You only provided the size of one dataset only, what about the others? (3) About the random dataset: how can you be sure that an object is present in the selected image regions? (4) How different toddler and adult datasets are? (5) What about the variability and complexity of objects? How a state-of-art object recognition method would perform on the datasets?

- Row 304: “…temporally adjacent image xt+∆T at time  t+∆T “. If this is the case, Delta t should be related to the video frame rate. In my opinion, a clearer formulation would consider t and t+1 instead, a more commonly adopted convention with videos. If I misunderstood, the motivations behind the choice of Delta t to be 1/30 should be given.

- Eq 1, unclear. If k is different from t, it would still be equal to t+Delta, which would contribute to both the numerator and denominator. Please clarify

- “We also observe that the gap between the Toddler fixation dataset and other datasets built with the ground truth about objects’ identities is small. We see a similar trend with the adult datasets. This suggests that a biologically inspired visual learning model like SimCLR-TT can leverage human gaze behaviours to build better visual representations. “ I miss the intuition behind this observation

- I find Fig. 4 a bit ambiguous. It may be useful to add details to favour the comprehension, e.g. what the colours refer to.

- If I’m correctly interpreting Figure 4, it seems to me that the difference between using toddlers and adults data is very little. How do you explain this fact?

- What’s the expected generalization of these observations? For instance: all the experiments are based on a fixed backbone, the ResNet18. What could be the impact of changing the representation?

- The conclusions in Sec, 4.2 are not surprising. If images are cropped so that the object is at the centre and with less background the recognition improves. This seems to be more a sanity check than a useful insight.

---

> ### Author Response · Authors · 2024-11-25
> **Response to Reviewer 74Gv [1/2]**
>
> We appreciate your careful review and the valuable comments you provided. Below are our detailed responses to each of your comments and questions.
>
> - **W1: Besides the scientific question, which I find very interesting indeed, it’s not clear to me what would be the use and the impact of this analysis.**
>
> Our analysis reveals how toddler's gaze behavior plays a role in learning object representations, which can help us understand early visual development. Our goal is not to improve conventional machine learning systems that learn passively by finding structure in predefined datasets.
>
> - **W2: What are the practical scenarios where these results can be exploited?**
>
> There may be long term advantages to building machine learning systems that learn more like humans, see, e.g., [1].
>
> [1] Sheybani, Saber, et al. "Curriculum learning with infant egocentric videos." Advances in Neural Information Processing Systems 36 (2024).
>
> - **W3: How the empirical observations that you are making can be turned into actionable insights? For instance, can the insights be turned into suggestions on the way datasets for object recognition should be collected?**
>
> We think it's important to highlight that humans do not learn by passively finding structure in predefined datasets as most artificial object recognition systems do. Instead they select their training inputs through their behavior based on their current state of knowledge. We hope our work will contribute to a paradigm shift in the machine learning community to embrace such active forms of learning.
>
> - **W4: However, in my opinion, the provided experiments are not enough to speak in favour of the generality of the outcomes (see Questions).**
>
> We have responded to your questions point by point as below.
>
> - **Q1: The contributions should be clearly stated in the introduction, to fully unveil the intentions of the authors. For instance, are the datasets a contribution? Will you share the data with the community?**
>
> Thank you for pointing this out. We now clearly state our contributions in the Introduction. Because there are faces of people visible in the videos, sharing of the data is impossible because of privacy concerns.
>
> - **Q2.1: Can you please explain more clearly the motivations behind the centroid fixation dataset? How can you be sure that an object is always present in the selected image portion?**
>
> (1) The centroid fixation dataset simulates fixed eyes (no eye movements, just head motion). This allows us to assess the importance of eye gaze behaviors for object learning.
>
> (2) There seems to be a misunderstanding here. Except for the two ''oracle'' datasets, none of the datasets are constructed to display objects in the selected image portion. Our objective is studying the importance of toddlers' gaze behaviors for learning object representations with temporal slowness. For this it is meaningful to compare toddler's actual gaze with simulated random eye movements or centroid fixation simulating no eye movements (but only head movement).
>
> - **Q2.2: You only provided the size of one dataset only, what about the others?**
>
> All the cropping datasets (toddler, adult, random, and centroid fixation datasets) contain 559,522 images. Objects fixation dataset and Plain background dataset contain 271,754 and 1,536 images, respectively. We report these numbers} on Lines 147, 210, and 214.
>
> - **Q2.3: About the random dataset: how can you be sure that an object is present in the selected image regions?**
>
> In the random fixation dataset, the cropping is done by uniformly random fixation points. Indeed, this is expected to lead to a low probability of consistently capturing objects within the selected image regions. Nevertheless, it is an important baseline to evaluate the effectiveness of actual human gaze strategies.
>
> - **Q2.4: How different toddler and adult datasets are?**
>
> The toddler and adult datasets differ primarily in the gaze behavior exhibited during the recordings. Please see Section 4.4. We used statistical tests to identify the key differences between the toddler and adult datasets. One key point is that toddlers spend more time observing objects while manipulating them compared to adults.
>
> - **Q2.5: What about the variability and complexity of objects? How a state-of-art object recognition method would perform on the datasets?**
>
> (1) The toddler fixation dataset comprises a diverse set of 24 toys, varying in shape, color, and function, providing a rich context for learning. We added Table 2 to show all the objects.
>
> (2) Please clarify your question. Do you mean a method that has been trained with supervision on a large dataset such as ImageNet? Or do you ask about the result of training a state-of-the-art self-supervised method on this dataset?

---

> ### Author Response · Authors · 2024-11-25
> **Response to Reviewer 74Gv [2/2]**
>
> - **Q3:  Row 304: “…temporally adjacent image xt+$\Delta{T}$ at time t+$\Delta{T}$ “. If this is the case, Delta t should be related to the video frame rate. In my opinion, a clearer formulation would consider t and t+1 instead, a more commonly adopted convention with videos. If I misunderstood, the motivations behind the choice of Delta t to be 1/30 should be given.**
>
> After some back and forth, we have decided to stick to the usage of $\Delta {T}$, but we have clarified how it relates to the frame rate: $\Delta {T}$ must be an integer multiple of the inverse frame rate.
>
> In addition, we replaced the word 'adjacent' with 'close' to provide a more precise description of the relationships between image pairs. Please check the details in Section 3.2 and Figure 3.
>
> - **Q4: Eq 1, unclear. If k is different from t, it would still be equal to t+$\Delta$, which would contribute to both the numerator and denominator. Please clarify.**
>
> Thanks for pointing out, $k$ can be equal to $t+\Delta{T}$. We clarify that in Line 242.
>
> - **Q5: “We also observe that the gap between the Toddler fixation dataset and other datasets built with the ground truth about objects’ identities is small. We see a similar trend with the adult datasets. This suggests that a biologically inspired visual learning model like SimCLR-TT can leverage human gaze behaviours to build better visual representations. “ I miss the intuition behind this observation.**
>
> Thank you for your feedback. We removed this statement.
>
> - **Q6: I find Fig. 4 a bit ambiguous. It may be useful to add details to favour the comprehension, e.g. what the colours refer to.**
>
> We expanded the caption of Figure 4 to clarify this.
>
> - **Q7: If I’m correctly interpreting Figure 4, it seems to me that the difference between using toddlers and adults data is very little. How do you explain this fact?**
>
> Differences in recognition accuracy between toddlers and adults are indeed small but consistent across models. In Section 4.3, we find that amplifying the temporal slowness of our representation enhances the difference between toddlers’ and adults’ representations. Section 4.4 highlights some visual behavioral differences between toddlers and adults through statistical testing.
>
> We also added Appendix B.4 to highlight some difference between toddlers' and adults' data.
>
> - **Q8: What’s the expected generalization of these observations? For instance: all the experiments are based on a fixed backbone, the ResNet18. What could be the impact of changing the representation?**
>
> To address any concerns regarding the generalization of our observations, we have added results based on ResNet50 in Appendix B.2. The results of BYOL-TT and SimCLR-TT both confirm our findings with ResNet18.
>
> - **Q9: The conclusions in Sec, 4.2 are not surprising. If images are cropped so that the object is at the centre and with less background the recognition improves. This seems to be more a sanity check than a useful insight.**
>
> The finding that cropping the images improves object recognition is intuitive. However, to the best of our knowledge, we are the first to make that clear when learning with temporal slowness on egocentric videos with the real gaze point.

---

> > ### Comment · Reviewer_74Gv · 2024-11-26
> >
> > I sincerely thank you for the effort in considering my review. I still have two concerns that I will try to clarify in the following, and I would like to discuss.
> >
> > Your answer about the contributions makes me reason about the appropriateness of the venue. In it, you emphasise what is the main outcome of your analysis, and what *is not* a contribution. Still, I believe that a clear statement on what *is* the contribution is missing.
> > In response to another reviewer who has similar concerns, you point out that “This work falls within the scope of ICLR as “applications to neuroscience and cognitive science.” However, you selected as Primary Area: unsupervised, self-supervised, semi-supervised, and supervised representation learning. This is misleading since one expects to find contributions in that direction.
> >
> > On the datasets, there might have been misunderstandings indeed. However, still, it’s not obvious to me the strength of the considerations that you can make. One important observation that may arise is that focusing on objects for some time (as toddlers do) improves the recognition abilities, which are also made more robust against view changes if a time analysis is applied. This seems very reasonable. However, adults also fix their gaze on the objects they are manipulating, even if for a shorter amount of time. This may explain why the performance of models with toddlers and adults data is not that different. Also, this may suggest that toddlers' data are more numerous than the adults'. What I still can not grasp is the significance of using a random and a centroid fixation dataset, where, for sure, the amount of images where the object is not visible (even for a single part) is higher than in the other cases. I understand that “none of the datasets is constructed to display objects in the selected image portion”, but the two datasets guided by the gaze position should naturally lead to “more focused” data. In this sense, these two datasets might be less descriptive of the objects, which may explain the change in performance. Does these interpretations make sense?

---

> ### Author Response · Authors · 2024-11-26
> **Response by Authors**
>
> We sincerely appreciate your thoughtful feedback and address your two main concerns below:
>
> **Regarding your first concern:**
>
> Clearly, our research is very interdisciplinary lying at the interface of Machine Learning and Developmental Science. In the latest version of the Introduction, we (positively) spell out our contributions as follows:
>
> ***
>
> In sum, our main contributions are:
>
> - We present the first ever study training SSL models on natural egocentric visual input derived from eye tracking in toddlers during play sessions.
>
> - We find that toddlers' gaze strategy improves the learning of invariant object representations compared to several baselines.
>
> - We show that toddlers' visual experience is more suitable for learning object representations through time-based SSL than adults'.
>
> ***
>
> We’re sorry if our work raised certain expectations with you that weren’t met. However, we would argue that our work falls 100% inside within “unsupervised, self-supervised, semi-supervised, and supervised representation learning”. That’s exactly what we do — it’s just that we feed the self-supervised learning algorithms with actual eye-tracking derived, first-person input from toddlers and adults, which has never been attempted before! We do so to answer questions about how human self-supervised representation learning might work, but that doesn’t change the fact that the work is about SSL. Is this mainstream Machine Learning research? Clearly not. But is it novel and falls within the scope of the conference? Clearly yes, we’d argue.
>
> **Regarding your second concern:**
>
> We agree that one factor contributing to the worse performance of the random fixation dataset could be that the training objects may be falling inside the random crops less often. However, Figure 6D suggests that overall looking time at the target objects is not very predictive of the quality of the learned representations. For the Centroid fixation dataset, it is quite unclear if there is a noteworthy reduction in looking at objects because eyes and head are usually well aligned during natural behavior. What our experiments clearly show, however, is that eye gaze matters and that has never been shown before.
>
> Please feel free to contact us with any further questions or comments. We'd be happy to discuss and clarify our work.

---

> > ### Comment · Reviewer_74Gv · 2024-12-02
> >
> > Thanks again to the authors for their detailed answers. The second concern has now been clarified.
> >
> > Regarding the contributions, I agree that there is a novelty. Still, I am not sure it is presented in a unified way. If, as answered to another reviewer, your primary interest "... is in understanding how toddlers learn object representations while interacting with their environment," then the point here is using ML (and SSL in particular) to support your hypothesis (in this sense it's more an "application" paper). If instead is to reason on how to appropriately collect data for gaining view-invariance by using SSL, then it would be more methodological. Maybe it's both of them, it's a matter of highlighting the goals so that they are immediately clear to the reader and the paper is correctly placed in the context of the conference.
> >
> > I enjoyed this discussion, and I'm happy to increase my rating.

---

> > > ### Author Response · Authors · 2024-12-02
> > > **Response by Authors**
> > >
> > > We sincerely appreciate your feedback and the increase in rating. We also enjoyed the discussion with you, and as we addressed each of your questions and concerns, we were able to continuously improve the quality of our work.
> > >
> > > Since our work is interdisciplinary, we have been careful in choosing our words when summarizing our contributions, aiming to present them as clearly as possible for both reviewers and potential readers. We fully understand your emphasis on clarity regarding the contributions of the work, and we will take extra care in describing the contributions and innovations in future interdisciplinary research.
> > >
> > > Finally, we are grateful for the detailed comments and questions you provided on our work and appreciate the time and effort you dedicated to the discussion.

---

### Official Review · Reviewer_YxBk · 2024-11-04

**Soundness:** 2
**Presentation:** 3
**Contribution:** 2
**Rating:** 6
**Confidence:** 4

**Summary:**

This manuscript starts by stating that toddlers are quick in learning view-invariant object recognition. In particular, the writing implies that toddlers are more efficient learners compared to existing self-supervised learning methods. The authors identify that the eye tracking data of toddlers can provide a semantically stable video dataset, which can be exploited by learning time-wise consistencies when coupled with appropriate cropping. They define 3 cropping strategies: gaze-based, random, centroid (reflecting head orientation), as well as 2 oracle methods (objects-focused and so-called “plain background”) then train a self-supervised-through-time method (SimCLR-TT), then perform linear evaluation (keep the ResNet-18 fixed, train 1 classifier layer). They show that when training on data curated using toddlers’ gaze, performance can get close to that of a model trained on objects-focused oracle data. Their ablation study shows that a field-of-view (crop size) of 128x128 is best, and that toddlers hold their gaze steady (shown by trying different time steps in training SimCLR-TT) compared to adults.

**Strengths:**

The paper is interesting to read, well-written in general, and can be followed reasonably easily. The manuscript poses the question regarding understanding view-invariant object recognition learning, then tackles it methodically by proposing pre-processing steps on the Bambach et al. 2018 dataset and designing several experiments based on it.

The experiments seem to affirm the authors’ assumptions in that:
- toddlers gaze differently at objects, compared to adults, especially in terms of how long they gaze at the objects that they are holding in their hands.
- eye gaze signals can be used for a type of self-supervised learning, thanks to the tendency for both toddlers and adults to look at the same object, over a few time steps.

**Weaknesses:**

A core thesis of this paper seems to be that existing SSL approaches could learn better. The first sentence of the conclusion says: “Current SSL approaches still struggle to learn robust human-like object representations and the reasons for this remain unclear.” - however, this is never backed by evidence neither in the related work nor in the experiments. It may seem that the authors pose this paper as a first step towards curating a large-scale toddler-driven dataset that would improve upon typical self-supervised learning datasets such as ImageNet-1K or JFT-300M, but this is hard to agree with. It would help if the authors provide specific evidence or citations supporting their claim about the limitations of current SSL approaches in learning human-like object representations.

Furthermore, the paper puts a lot of effort into differentiating between the behavior of toddlers and adults. Some observations are intuitive (toddlers fixate longer at the objects that they hold in their hands), but others are less so (no real difference in SSL results in Figure 4). I can see the value of this paper as a behavioral science paper - one that uses machine learning to show how toddlers may learn view-invariant object recognition. This is because many results seem to be revealing toddlers’ behavioral patterns rather than contributing to improving machine learning. Could the authors please clarify what the primary contribution of their paper is? Is the paper intended to advance machine learning techniques, or is it primarily a study of toddler behavior using machine learning as a tool?

**Questions:**

- Are you the first to apply SimCLR-TT (or another self-supervised learning through time work) to toddler videos? What about works like [1]?
- Did you consider evaluating several SSL methods such as done in [2]? What was the reason for using a SimCLR-based method? Many SSL methods have been introduced since 2020 (e.g. BYOL, SimSiam, DINO, MAE) and many of them can be easily adapted to the SSL-through-time setting.
- In FIgure 1, why are the examples in E and F of an object that does not exist in the egocentric video? This confuses me somewhat, because I assumed B-E are all different ways of cropping the original A.

[1] Aubret, Arthur, Céline Teulièr, and Jochen Triesch. "Toddler-inspired embodied vision for learning object representations." 2022 IEEE International Conference on Development and Learning (ICDL). IEEE, 2022.

[2] Pandey, Lalit, Samantha Wood, and Justin Wood. "Are vision transformers more data hungry than newborn visual systems?." Advances in Neural Information Processing Systems 36 (2024).

---

> ### Author Response · Authors · 2024-11-25
> **Response to Reviewer YxBk**
>
> Thank you for the careful and very helpful analysis of our manuscript. Please find the detailed responses below.
>
> - **Weakness: Clarifying the contribution and claims about SSL limitations.**
>
> We agree that our motivation and contributions were not presented sufficiently clearly. We have fixed that in the revised version.
>
> Our primary interest is in understanding how toddlers learn object representations while interacting with their environment rather than proposing improved machine learning methods.
>
> We use machine learning as a tool to analyze how toddlers' gaze behavior may contribute to their learning of object representations. This work falls within the scope of ICLR as “applications to neuroscience and cognitive science.” We have adjusted the title, abstract, and introduction to clarify this perspective, and we also highlight our contributions point by point in the revised introduction.
>
> Regarding the limitations of current SSL approaches, studies [1-3] have confirmed that current SSL methods still struggle to learn robust human-like object representations. This is mentioned in the Introduction in Lines 036-039.
>
> [1] Dong, Yinpeng, et al. "Viewfool: Evaluating the robustness of visual recognition to adversarial viewpoints." Advances in Neural Information Processing Systems 35 (2022): 36789-36803.
>
> [2] Abbas, Amro, and Stéphane Deny. "Progress and limitations of deep networks to recognize objects in unusual poses." Proceedings of the AAAI Conference on Artificial Intelligence. Vol. 37. No. 1. 2023.
>
> [3] Ruan, Shouwei, et al. "Towards viewpoint-invariant visual recognition via adversarial training." Proceedings of the IEEE/CVF International Conference on Computer Vision. 2023.
>
> - **Q1: Are you the first to apply SimCLR-TT (or another self-supervised learning through time work) to toddler videos? What about works like [1]?**
>
> We are not the first to apply time-based SSL to first-person videos, as acknowledged in Section 2 (Lines 106-107). However, to the best of our knowledge we are the first to exploit eye tracking-derived precise gaze information. This is an essential advance given the highly foveated nature of human vision that strongly biases the sampling of information to the (rapidly changing!) gaze location.
>
> - **Q2: Did you consider evaluating several SSL methods such as done in [2]? What was the reason for using a SimCLR-based method? Many SSL methods have been introduced since 2020 (e.g. BYOL, SimSiam, DINO, MAE) and many of them can be easily adapted to the SSL through-time setting.**
>
> We have now included additional results using BYOL-TT and also larger networks in Appendix B.1 and B.2, which confirm our conclusions. The reason for focusing on SimCLR-TT is that it has been widely used in previous studies [4-8].
>
> [4] Schneider, Felix, et al. "Contrastive learning through time." SVRHM 2021 Workshop@ NeurIPS. 2021.
>
> [5] Sheybani, Saber, et al. "Curriculum learning with infant egocentric videos." Advances in Neural Information Processing Systems 36 (2024).
>
> [6] Aubret, Arthur, Céline Teulièr, and Jochen Triesch. "Toddler-inspired embodied vision for learning object representations." 2022 IEEE International Conference on Development and Learning (ICDL). IEEE, 2022.
>
> [7] Pandey, Lalit, Samantha Wood, and Justin Wood. "Are vision transformers more data hungry than newborn visual systems?." Advances in Neural Information Processing Systems 36 (2024).
>
> [8] Aubret, Arthur, et al. "Time to augment self-supervised visual representation learning." arXiv preprint arXiv:2207.13492 (2022).
>
> - **Q3: In Figure 1, why are the examples in E and F of an object that does not exist in the egocentric video? This confuses me somewhat, because I assumed B-E are all different ways of cropping the original A.**
>
> Examples E and F also represent objects present in the egocentric videos, but they are not cropped from subsequent frames of the videos as in B-D. We have now added example images and labels for the 24 toys in Appendix C. We also replaced the object images and added white margins in E and F from Figure 1 to avoid any potential confusion.

---

> > ### Comment · Reviewer_YxBk · 2024-12-01
> >
> > Thank you very much for providing a detailed response to my review. I generally appreciated the responses that the authors provided to the other reviews too.
> >
> > Given that the main insight is on toddler behavior (rather than strong and direct suggestions on a paradigm shift for SSL itself) and that the manuscript revisions support this, I am happy to increase my rating.
> >
> > While some of the experimental results may seem obvious (the design of the baselines may naturally lead to the measured performance differences), such a confirmation is still valuable to gain insights toward (a) understanding toddlers' learning behavior and (b) considering alternative approaches to designing SSL methods. In my opinion, the authors have generally shown sufficient rigor in the design, execution, and presentation of their experiments.

---

> ### Author Response · Authors · 2024-12-01
> **Response by Authors**
>
> We sincerely appreciate your feedback on our rebuttal and the decision to increase your rating. Your insights, particularly regarding the clarity of our manuscript and the experimental aspects, have significantly strengthened our work.
>
> As you mentioned, analyzing toddler behavior alongside their visual input is crucial for developing more robust representations. In the future, we plan to incorporate additional bio-inspired mechanisms and conduct more comprehensive experiments.
>
> Thank you for your time and valuable contributions during the review. Your input has greatly helped us improve the quality of our manuscript.

---

### Author Response · Authors · 2024-11-26
**Official Comment by Authors**

We would like to thank all the reviewers for taking the time to read our paper and provide valuable questions and comments. We carefully considered the feedback and revised the manuscript accordingly. We summarize the changes we have made:

(1) We have revised the title, abstract, and introduction to better clarify the contributions of our work.

(2) We updated Figure 1 for clarification.

(3) We refined the description of Eq. 1 to make it more precise and clear.

(4) Regarding the new experiment:

  - **Appendix B.1:** Introduced a new SSL method, BYOL-TT.
  - **Appendix B.2:** Compared the performance of ResNet50 and ResNet18 encoders on SimCLR-TT and BYOL-TT.
  - **Appendix B.3:** Discussed the impact of various sampling methods used to balance the dataset.
  - **Appendix B.4:** Highlighted the differences between toddlers and adults.

(5) Table 2 in Appendix C displays examples of every object used in the experiment.

(6) Some minor revisions were made to improve grammar and writing style.


We hope the revised manuscript addresses most of the concerns raised. If you agree, we kindly request that you re-evaluate our paper based on this updated version. We are also happy to discuss and clarify any remaining or new issues.

Once again, we sincerely thank you for your contributions to improving the quality of our work.

---

### Author Response · Authors · 2024-12-04
**Thanks to all reviewers**

We sincerely appreciate the time and effort all reviewers have dedicated throughout both the review and rebuttal phases.

- We addressed the concerns Reviewer **YxBk** and Reviewer **74Gv** raised. We are pleased that they found the discussion engaging and increased their ratings. We are also grateful for their positive feedback, particularly the suggestion about the contribution.

- We hope the revised manuscript clarifies the concerns Reviewer **6nrm** and Reviewer **BEUM** raised. We would appreciate it if you could re-evaluate our updated manuscript.

Finally, we thank all the reviewers again for their valuable comments and feedback, which have significantly enhanced the quality of this work.

---

### Note · Authors · 2025-02-12

I have read and agree with the venue's withdrawal policy on behalf of myself and my co-authors.

---

### Meta-Review · Area_Chair_qDqJ · 2024-12-17

**Metareview:**

The submission initially had mixed reviews, and the major issues were:

1. overclaims about limitations of current SSL approaches in learning human-like object representations [YxBk]
2. what is the primary contribution: to advance ML techniques, or as a study of toddler behavior using ML as a tool? Suitability of the venue? [YxBk, 74Gv]
3. Try other SSL methods [YxBk, BEUM]
4. What is the practical implication (to ML)? [74Gv]
5. questions about generality of outcomes [74Gv , 6nrm]
6. add standard data augmentations [6nrm]
7. insights about what is learned: what features are affected by toddler gaze? differences in feature maps between adults/toddlers? [6nrm]
8. are differences between todder/adults significant? [6nrm]
9. comparison with random/centroid fixations do not contain objects of specific categories [BEUM]
10. no significant difference in experiment results between adults/toddlers [BEUM]

The authors wrote a response, and there were good and useful discussions during the author-reviewer period. Reviewers YxBk and 74Gv raised a concern about whether the primary contribution is to advance ML or to study toddler behavior using ML (Point 2). The authors replied:
- We use machine learning as a tool to analyze how toddlers' gaze behavior may contribute to their learning of object representations. This work falls within the scope of ICLR as “applications to neuroscience and cognitive science.” We have adjusted the title, abstract, and introduction to clarify this perspective, and we also highlight our contributions point by point in the revised introduction.  In sum, our main contributions are:
  1) We present the first ever study training SSL models on natural egocentric visual input derived from eye tracking in toddlers during play sessions
  2) We find that toddlers' gaze strategy improves the learning of invariant object representations compared to several baselines.
  3) We show that toddlers' visual experience is more suitable for learning object representations through time-based SSL than adults'.

The reviewers appreciated the interdisciplinary work. The AC agree with the authors and reviewers, that the paper generally fits into the scope of ICLR as an interdisciplinary paper on "applications to cognitive science".

However, there are still several major concerns raised by Reviewers 6nrm and BEUM that were not addressed well:
- Point 10: The premise of the paper rests on showing a difference between performance of models learned from toddler or adult fixations. However, this does not appear to be statistically significant (Fig 4 or Table 1 (bold result)). Indeed there are only 3 trials here and the standard deviation is on par or larger than the accuracy difference, so this result is not very convincing. When asked about this by BEUM, authors only responded that "The differences between toddlers and adults are indeed small (albeit consistent across network models)," and did not offer any statistical test to back up this statement.
- Point 9: toddler/adult fixations are biased by the appearance of the object, but random fixations are on the whole image. So it is obvious that toddler/adult fixations will be getter.  Perhaps a more interesting baseline is to place random fixations on the object.
- Point 7: more analysis of the differences in features/representations between the two models would be interesting, especially for a ML conference since these insights could suggest potential avenues for algorithm design later.

Finally, the AC considered another issue about the motivation. The paper focuses on toddler's eye gaze, and posits that their gaze behavior better supports learning object representations, compared to adult gaze behavior. However, there are some nuances here that could be discussed and controlled for. The toddlers are in the process of learning (about objects), while adults have already learned them. So we may expect toddler fixations to support learning, while adults fixations are mainly supporting fast inference. Similarly, the toddler is actively engaging the object, while the adult is passively viewing. So, it is not necessarily the toddler that is important, but the engaged learning process. In this sense, the same experiment could be run using adults learning novel objects (e.g., see tarrlab stimuli https://sites.google.com/andrew.cmu.edu/tarrlab/stimuli) vs adults recognizing those objects.

Overall, the paper in its current form is not quite ready for a top-tier ML conference, especially with regards to Points 10 and 7. Thus, the AC recommends reject, and recommends the authors to carefully revise their paper and resubmit it.

**Additional Comments On Reviewer Discussion:**

see above

---

### Decision · Program_Chairs · 2025-01-22

Reject